Recognizing IgA-class endomysial antibody equivalent binding patterns on monkey liver substrate through EfficientNet architectures and deep learning

http://orcid.org/0000-0002-9145-1506 Soylu Mehmet 1
http://orcid.org/0000-0003-4305-7800 Bozkir Ahmet Selman 2 selman@cs.hacettepe.edu.tr
1 Department of Medical Microbiology, Ege University , İzmir , Turkey
2 Department of Computer Engineering, Hacettepe University , Ankara , Turkey
Sergi Consolato
Electronic publication date: 2025 Oct 15
Publication date: 2025
Volume: 13
Electronic Location ID: e20191
Received 2025 May 5; Accepted 2025 Sep 15
Copyright: © 2025 Soylu and Bozkir
Copyright year: 2025
Copyright holder: Soylu and Bozkir
License: This is an open access article distributed under the terms of the Creative Commons Attribution License, which permits unrestricted use, distribution, reproduction and adaptation in any medium and for any purpose provided that it is properly attributed. For attribution, the original author(s), title, publication source (PeerJ) and either DOI or URL of the article must be cited.
License URL: https://creativecommons.org/licenses/by/4.0/

Keywords: Machine learning, Computer vision, Celiac diagnosis, IgA endomysial antibody tests

Funding: The authors received no funding for this work.

==============================
Deep learning offers promising potential for automating the interpretation of immunoglobulin A (IgA) endomysial antibody (EMA) tests, a critical serological test for the diagnosis of celiac disease that currently requires labor-intensive and subjective human interpretation. In this study, we employ and comprehensively evaluate the performance of the EfficientNet and EfficientNetV2 architectures in binary (positive vs negative, where all weak and strong positive signals were grouped as positive), three-class (negative, weak positive, strong positive), and four-class (negative, weak positive, strong positive and gray zone) classification scenarios using immunofluorescence images of IgA EMA equivalent (EMA-eq) tests. Our experiments on 368 clinical samples show high performance, with EfficientNetV2-S achieving an accuracy of 99.37% in binary classification, 95.28% in three-class classification, and 86.98% in the complex four-class scenario that introduces gray zone cases as a distinct interpretive category. Contrary to conventional assumptions, medium-sized deep architectures consistently outperformed their larger counterparts. The superior performance of the EfficientNet-V2 models can be attributed to their architectural innovations and higher input resolution (640 × 640 pixels), which proved critical for capturing subtle immunofluorescence patterns. We also incorporate HiRes-CAM (Class Activation Mapping), a convolutional neural network oriented visual explanation tool, to better understand the decisions of the underlying trained deep learning model in an explainable artificial intelligence (AI) manner. This study demonstrates that deep learning has the potential to achieve expert-level performance in EMA-eq test interpretation, offering a path toward more standardized, efficient and objective celiac disease diagnosis while reducing the burden on specialist medical staff.

Introduction

Celiac disease (CD) is a prevalent autoimmune disorder triggered by gluten ingestion, affecting approximately 1% of the global population (Singh et al., 2018). The clinical presentation of CD is highly heterogeneous, ranging from gastrointestinal symptoms like diarrhea and abdominal pain to extra-intestinal manifestations such as dermatitis herpetiformis and neurological disorders (Guandalini & Assiri, 2014). This wide spectrum of symptoms often leads to delayed or missed diagnoses, contributing to the underrecognition of CD worldwide (Rostami et al., 1999). Studies have reported that the average delay between symptom onset and diagnosis can range from 6 to 10 years, with some populations experiencing even longer diagnostic delays (Green & Jabri, 2003; Rubio-Tapia et al., 2012). Misdiagnosis or late diagnosis can lead to irreversible complications such as osteoporosis, infertility, neurological deficits, and even increased malignancy risk. Economically, delayed diagnosis contributes to increased healthcare utilization, including unnecessary tests and specialist consultations, resulting in substantial financial burden for both patients and healthcare systems. Recent European studies estimated that undiagnosed celiac disease can result in excess healthcare costs (Fuchs et al., 2018; Bokemeyer et al., 2025). Furthermore, subjective interpretation of EMA test results contributes to inter-observer variability, increasing the risk of diagnostic malpractice and limiting the scalability of endomysial antibody (EMA) testing in broader clinical settings.

Traditionally, the diagnosis of CD has relied on small bowel biopsies demonstrating villous atrophy, intraepithelial lymphocytosis, and crypt hyperplasia (Catassi & Fasano, 2008). However, the invasive nature of biopsies and potential sampling errors have prompted the exploration of non-invasive diagnostic tools. Serological tests for CD-specific immunoglobulin A (IgA) antibodies, particularly EMA and anti-tissue transglutaminase antibody (anti-tTG), have emerged as valuable alternatives due to their high sensitivity and specificity (Sheppard et al., 2022). Recent guidelines from the European Society for Pediatric Gastroenterology, Hepatology and Nutrition (ESPGHAN) suggest that a diagnosis of CD can be established without biopsy in symptomatic children with elevated Anti-TG levels and positive EMA test (Husby et al., 2020). This non-invasive approach has also shown promise in adult populations, highlighting the growing importance of EMA testing in CD diagnosis. The EMA test is performed using an indirect immunofluorescence assay (IIF), in which patient serum is incubated on tissue sections—commonly monkey esophagus, human umbilical cord, also EMA-eq tests can be performed using IIF with monkey liver sections (Wolf et al., 2016). If IgA antibodies are present, they bind to antigens along the connective tissue framework, producing characteristic fluorescence patterns upon the application of a labeled anti-IgA conjugate. On monkey esophagus, this reactivity localizes to the connective tissue layers surrounding smooth muscle fibers of the lamina muscularis mucosae and tunica muscularis, which are rich in tissue transglutaminase, whereas on monkey liver substrate, binding predominantly occurs in the walls of the sinusoidal vessels (Wolf et al., 2016; Schauer et al., 2023). Despite its high sensitivity, specificity and clinical value, particularly when combined with anti-tTG IgA testing, EMA interpretation remains resource-intensive, time-consuming, and prone to inter-observer variability (Murray, Frey & Oliva-Hemker, 2018; Shahmirzadi & Sohrabi, 2019; Anbardar et al., 2023). These operational challenges limit its scalability in routine clinical practice, especially in settings lacking experienced personnel. Computer vision, on the other hand, has seen exponential growth in recent years, driven by advances in data availability and computational power. As Sana et al. (2020) noted, deep learning approaches are now widely used in medical image analysis for screening, diagnosis, and clinical decision support, offering potential solutions to the limitations of manual test interpretation. Interpreting immunofluorescence-based serological tests such as the EMA assay remains a visually complex and expertise-dependent task, particularly in borderline or low-intensity cases. Inspired by recent successes of deep learning in histopathological image classification—where convolutional neural networks (CNNs) have rivaled or surpassed human experts in recognizing subtle phenotypic features—this study explores whether CNN-based models can similarly automate and standardize EMA test interpretation (Syed et al., 2019; Caetano dos Santos et al., 2019).

This research aims to investigate (i) whether supervised deep learning algorithms can be trained to classify IgA EMA-eq immunofluorescence patterns with accuracy comparable to expert judgment and (ii) whether these models can reliably handle ambiguous “gray zone” cases that frequently arise in clinical practice. This study addresses a significant diagnostic challenge in celiac disease screening by focusing on monkey liver–based celiac autoantibody binding images, which can be regarded as diagnostically equivalent to EMA detected on classic muscle substrates such as monkey esophagus or human umbilical cord images, which differ from the more commonly studied jejunal biopsy or endoscopy images. To our knowledge, this is the first study to evaluate the feasibility and performance of EfficientNet and EfficientNetV2 deep learning architectures on such immunofluorescence images, using both standard and gray-zone-inclusive classification schemes. Specifically, this study makes several important contributions as follows: We perform a comprehensive evaluation of multiple EfficientNet and EfficientNetV2 models in the problem domain.

We show that deep learning methods can reach the level of an expert, with 99.37% accuracy in identifying positive and negative samples.

We employ HiRes-CAM (Class Activation Mapping) for the first time in this field to provide visual explanations for model predictions, enhancing interpretability for clinical users.

We release a curated and expert-annotated image dataset to facilitate further research and benchmarking in this domain.

The rest of this study is organized as follows. ‘Related Work’ reviews related work and background. ‘Materials and Methods’ details our materials and methods, including data acquisition stages and the deep learning models. ‘Experimental design’ presents experimental design, whereas ‘Results’ demonstrates the results of the experiments together with outcomes of the comparative study. ‘Discussion’ explores the implications and limitations from diverse perspectives. Finally, ‘Conclusions’ concludes with key contributions and future directions.

Related work

Recent advances in medical image classification have demonstrated the effectiveness of combining deep learning architectures with active learning strategies. Several studies have evaluated the utility of more recent CNN architectures in various domains. For example, Ali et al. (2022) reported that EfficientNet-B4 achieved 87. 91% precision for the classification of skin cancer, outperforming many traditional CNN models. In a study specific to celiac disease, Wei et al. (2019) used ResNet-50 and achieved class-specific precisions of 95.3%, 91%, and 89.2%, with area under curve (AUC) values exceeding 0.95.

Considering our problem domain, the use of convolutional neural networks (CNNs) in the diagnosis of autoimmune diseases has seen significant advancements. Specifically, for HEp-2 cell image classification, several state-of-the-art deep learning algorithms have been developed and are continually evolving. Gao et al. (2016) proposed a framework that learns hierarchical feature representations directly from raw cell images, eliminating the need for hand-crafted features. Their work emphasized the importance of factors such as data augmentation and image masks, demonstrating that rotating training images enhances model robustness and that using whole cell images, including backgrounds, yields better performance. They also highlighted the adaptability of CNN models across different datasets, outperforming traditional models like Bag-of-Features (BoF) and Fisher Vector (FV) models.

Further advancing the field, Rodrigues, Naldi & Mari (2020) conducted an extensive study on HEp-2 cell image classification using several CNN architectures, including LeNet-5, AlexNet, Inception-V3, VGG-16 and ResNet-50. Their findings indicated that Inception-V3, trained from scratch with data augmentation, achieved the highest accuracy (98.28%). Their study underscored the efficacy of raw images and data augmentation in improving model performance by increasing the diversity of the training dataset. More recently, Vununu, Lee & Kwon (2021) proposed a method that combines active learning and cross-modal transfer learning to classify HEp-2 cell images. This approach aimed to minimize manual annotation by using a small, annotated dataset to pre-train deep residual networks, which were then fine-tuned on a larger dataset using active learning techniques. Their method demonstrated high discrimination performance with minimal annotated data, simplifying the labeling process while maintaining accuracy.

Recent studies have also explored the application of CNNs in the histopathological classification of celiac disease. Carreras (2024) employed a ResNet-18 model on a large dataset of H&E-stained duodenal biopsy images reporting high accuracy for coeliac detection. Although Grad-CAM was used for interpretability, their work was restricted to biopsy images without addressing serological modalities. Similarly, Suchỳ, Vranay & Magyar (2025) compared multiple architectures such as residual network 18 (ResNet18), visual geometry group 16 (VGG16) and capsule networks on duodenal histopathology image patches. Their best-performing ResNet18 model, supported by balanced sampling and data augmentation, achieved over 90% accuracy but remained limited to tissue-based data. Denholm et al. (2022) implemented a multiple-instance learning approach on duodenal whole slide images (WSI) to detect celiac disease, surpassing an accuracy of 96% with external validation across scanners and staining batches. Jaeckle et al. (2025) demonstrated that a CNN-based model could reach pathologist-level performance (AUC > 0.99) on a multi-institutional dataset of over 3,000 duodenal biopsy images. These studies highlight the increasing maturity of AI applications in biopsy-based celiac diagnostics. Also Stoleru, Dulf & Ciobanu (2022) explored capsule endoscopy image sequences using classical image processing combined with machine learning classifiers, reporting high accuracy (94.1%) and F1-scores in detecting villous atrophy from video frames. However, deep CNN models remain underutilized in this modality. Furthermore, a recent systematic review by Sharif et al. (2023) examined eight original studies applying deep learning to celiac disease diagnosis using endoscopic or capsule endoscopy images. Various CNN-based methods such as GoogLeNet, ResNet50, and Inception-v3 achieved diagnostic accuracies between 84% and 100%. However, their review identified significant limitations in the existing literature, including small sample sizes, a paucity of dataset diversity, and challenges in detecting mild mucosal lesions (e.g., Marsh I–II). Notably, none of the mentioned studies employed serological imaging or explainable AI techniques. In contrast, our study using EfficientNet-based models and HiRes-CAM visualization focuses on classifying EMA immunofluorescence images, a serological marker with clinical relevance in non-biopsy-based diagnostics. By leveraging interpretability methods and evaluating gray zone samples, we address a less explored but clinically significant domain in celiac disease diagnosis. Our work directly addresses these gaps by focusing on the supervised classification of EMA-eq images derived from monkey liver tissue, incorporating pixel-level visualization, benchmarking modern CNN architectures, and contributing a curated, labeled dataset for future research. Building on this foundation, Caetano dos Santos et al. (2019) explored machine learning techniques for diagnosing celiac disease. They developed an automated method for classifying IgA-class EMA test images from human umbilical cord using support vector machines (SVMs) and adaptive boosting (AdaBoost) on a dataset of 2,597 high-quality EMA images. Their model achieved high sensitivity (82.84%) and specificity (99.40%), highlighting the potential of machine learning in providing rapid and accurate analysis of EMA tests. They addressed common challenges in medical image classification, such as rotation invariance and class imbalance, employing techniques like AdaBoost ensemble learning and 10-fold cross-validation to enhance their model performance.

Materials and Methods

In this section, we first introduce our curated dataset. Next, we briefly explain the architectures of the EfficientNet family of convolutional neural networks (CNNs) used throughout the study, outlining their advantages and explaining why we chose them for this study. We then present the Evaluation metrics used to quantify the performance of the models we trained. Finally, in Experimental Design, we explain in detail our model development and experimentation.

Data gathering

This study included the analysis of 368 peripheral blood samples referred to the Laboratory of Medical Microbiology, Serology and Immunology, Ege University Faculty of Medicine Hospital with a preliminary diagnosis of celiac disease between June and December 2023. Of the evaluated cases, 197 (67.0%) were female and 97 (33.0%) were male. The age distribution showed that 153 patients (52.0%) were children under the age of 18, while 141 (48.0%) were adults. The mean age of the study group was 22.7 years, with a median of 16 years (range: 2 to 83 years), and an interquartile range (IQR) of 21.8 years. The study was conducted with the ethical approval obtained from the Ethics Committee of Ege University Faculty of Medicine (Approval No: 2023-1750/23-11.2T/13). The research was designed and conducted under ethical principles, including obtaining informed consent. This retrospective analysis used anonymized patient data and archived immunofluorescence images. Written informed consent was obtained at the time of initial clinical evaluation and the institutional ethics committee approved the study protocol. The dataset used in this study will be publicly available to researchers.

Data preprocessing

Initially, the samples were subjected to a centrifugal process at 3,000 g for 15 min, with the objective of effectuating the separation of the serum phase. Thereafter, the serum phase was transferred to secondary tubes. Serum samples are analyzed on the same day, unless they cannot be processed on the day of arrival, in which case they are stored at −20 ∘C. Serum samples were then analyzed through the EuroimmunTM EMA IgA kit on the EuroimmunTM Sprinter XL automated immunofluorescence microscopy slide preparation system. In this study, monkey liver tissue was used as the substrate for indirect immunofluorescence microscopy in IgA EMA-eq testing. The observed staining pattern did not localize to hepatocytes or bile ducts but rather delineated the extracellular matrix surrounding the hepatic sinusoids. Specifically, the fluorescence highlighted the reticulin-rich connective tissue network of the hepatic lobule, particularly within the perisinusoidal and perivascular regions. This microanatomical framework is known to express tissue transglutaminase (tTG), the principal autoantigen targeted by EMA in celiac disease. The resulting immunofluorescent signal appeared as a fine, thread-like pattern tracing the sinusoidal architecture, reflecting the localization of tTG within the stromal scaffold. This pattern, originally described as reticulin antibody binding (Seah et al., 1971), is considered analogous to the endomysial staining seen in striated or smooth muscle substrates and supports the use of monkey liver as a valid alternative for celiac autoantibody detection, and thus for classic EMA testing. Given the different image appearances, we shall call the positive celiac pattern on liver as EMA equivalent (EMA-eq), clearly distinguishing it from positivity on muscle endomysium.

Following this process, IIF preparations were obtained (Wolf et al., 2016; Schauer et al., 2023). Anti-TTG IgA levels were determined using a commercially available ELISA kit Euroimmun Anti-Tissue Transglutaminase IgA Order number: EA 1910-9601-A (Euroimmun AG, Lübeck, Germany), following the manufacturer’s instructions. The assay’s cut-off for positivity was ≥20 RU/mL.

All stained liver slides were scanned and digitized using the EuroimmunTM EUROPattern Microscope system, which provides a fully automated image acquisition workflow. The microscope is equipped with a motorized stage, autofocus, and standardized exposure controls, enabling consistent high-resolution fluorescence imaging without manual intervention during the acquisition process. For each serum specimen, the system captured a single standardized field of view, yielding one digital image per sample. In total, 368 original, untransformed images were obtained and subsequently used for training and evaluation. Importantly, the system does not perform interpretive analysis; all diagnostic classification was carried out manually by trained experts. Each image was independently reviewed by two laboratory personnel with over 10 years of experience in autoimmune serology. Only cases where both evaluators agreed were included in the final dataset. In cases of disagreement, the sample was excluded from the training and evaluation datasets.

To support the development of a standardized visual reference system, a subset of serum samples was subjected to serial dilutions at predefined ratios of 1:10, 1:32, 1:100, 1:320, and 1:1,000. Based on these dilutions, fluorescence signal patterns were interpreted as follows: samples negative at 1:10 were classified as negative; positive at 1:10 but negative at 1:32 were designated as gray zone; positive at 1:32 but negative at 1:100 were categorized as +1; those remaining positive at 1:100 but negative at 1:320 as +2; positive at 1:320 but negative at 1:1,000 as +3; and those exhibiting fluorescence at or beyond 1:1,000 were considered +4. This stratified titration protocol enabled the creation of a 30-image reference set, comprising five representative examples for each of the six intensity categories. Before the main assessment, all experts jointly reviewed this panel to calibrate their interpretation criteria. For the remaining samples, serial dilutions were not routinely performed; instead, fluorescence intensities were interpreted by directly matching each image to the standardized reference examples. This strategy reduced analysis time while promoting consistency and robustness across evaluations. To avoid classification bias due to undetectable antibody levels, patients with selective IgA deficiency were excluded from the dataset.

In total, three classification schemes were applied to the EMA-eq image dataset to evaluate the model under varying diagnostic complexities. First, in the EMA-eq 2-Class (i.e., binary) configuration, all samples showing any positive signal (1+ to 4+) were grouped as positive and contrasted with negative samples. Second, the EMA-eq 3-Class scheme maintained negative samples as a separate group, classified 1+ and 2+ as weak positives, and 3+ and 4+ as strong positives. Third, in the most detailed setting, the EMA-eq 4-class scheme introduced the gray zone as an independent category, alongside negative, weak positive, and strong positive samples. This multi-layered scoring method facilitated a comprehensive evaluation and enabled in-depth analysis of borderline and low-intensity fluorescence patterns. We presented sample images of each class in Fig. 1.

Figure 1 Sample images for each class in our corpus.

The number of training and validation samples for each category in all three classification datasets is detailed in Table 1. It is important to note that a 5-fold cross-validation scheme was implemented during the training and evaluation process. Therefore, the rows displaying numerical values for the “Overall” category denote the aggregate number of samples allocated for each fold’s training and validation partitions. Conversely, the rows situated beneath the “Overall” category illustrate the class-specific average sample distribution for each individual fold.

Table 1 Classwise number of samples per each dataset used in the study.

Dataset name	Class	Training	Validation	Total	Percentage (%)	
EMA-eq 2-Class	Overall	254	64	318	100	
	Positive (Strong + Weak)	134	35	169	53.1	
	Negative	120	29	149	46.8	
EMA-eq 3-Class	Overall	254	64	318	100	
	Strong positive	68	17	85	26.7	
	Weak positive	67	17	84	26.4	
	Negative	119	30	149	46.8	
EMA-eq 4-Class	Overall	294	74	368	100	
	Strong positive	68	17	85	23.0	
	Weak positive	67	17	84	22.8	
	Negative	119	30	149	40.4	
	Gray-zone	40	10	50	13.5	

EfficientNet

For years, researchers focused on improving accuracy by arbitrarily scaling networks along a single dimension—either increasing depth (more layers), width (more channels), or input resolution. However, this approach often resulted in diminishing returns, with massive computational costs yielding only marginal improvements in performance. In this regard, EfficientNet (Tan & Le, 2019) introduces a revolutionary approach to scaling convolutional neural networks that addresses a fundamental oversight in previous architectures like ResNet (He et al., 2016), VGG (Simonyan & Zisserman, 2014), densely convolutional networks (DenseNets) (Huang et al., 2017).

The fundamental insight behind EfficientNet is the empirical observations showing that network dimensions are highly interdependent. In fact, when input resolution increases, networks require greater depth to capture larger receptive fields and increased width to detect more fine-grained patterns. This observation led the researchers to propose a compound scaling method that systematically balances all three dimensions (i.e., depth, width and input resolution) simultaneously, rather than treating them as independent factors. EfficientNet formalizes this relationship using a compound coefficient ϕ that uniformly scales depth by αϕ, width by βϕ, and resolution by γϕ, while maintaining the constraint α⋅β2⋅γ2≈2 to ensure controlled computational growth. The constants α, β, and γ are determined through a small grid search on a baseline network, allowing the approach to adapt to different architectural foundations. The implementation process involves two steps: (a) establishing a baseline network (EfficientNet-B0) through neural architecture search that optimizes both accuracy and floating point operations per second (FLOPS) then (b) applying the compound scaling method to create a family of increasingly powerful models (B1–B7). This approach maintains the architectural benefits of the baseline while systematically scaling network capacity.

Beyond its practical benefits, EfficientNet represents a paradigm shift in network design philosophy, demonstrating that a thoughtful balance between architectural dimensions is more important than simply increasing model size. This insight not only improves the performance-efficiency trade-off, but also enables the deployment of powerful models in resource-constrained environments, making advanced computer vision capabilities more widely accessible. We chose EfficientNet as one of our main neural architectures because of its high scalability and ability to support different input resolutions.

EfficientNet-V2

Proposed by Tan & Le (2021), EfficientNet-V2, addresses critical limitations of the original EfficientNet architecture while significantly improving training speed and parameter efficiency. Where EfficientNet scaled uniformly across all network stages and struggled with large image sizes, EfficientNet-V2 introduces three key innovations.

Firstly, EfficientNet-V2 employs a training-aware neural architecture search that strategically combines MBConv blocks from the original model with new Fused-MBConv blocks in early layers. This hybrid approach has been shown to reduce the training bottlenecks caused by depthwise convolutions, which were particularly inefficient in EfficientNet’s initial stages. The architecture also employs smaller expansion ratios and more focused 3 × 3 kernels compared to EfficientNet’s varied kernel sizes. Secondly, EfficientNet-V2 implements a non-uniform scaling strategy that preferentially adds more layers to later stages of the network, contrasting with EfficientNet’s equal scaling across all stages. Furthermore, it restricts maximum image size to circumvent the memory constraints that plagued EfficientNet training. Thirdly, the model introduces adaptive progressive learning, where both image size and regularization strength increase throughout the training process. In contrast to the straightforward progressive resizing employed in EfficientNet, this approach employs a dynamic adjustment of regularization to align with the network capacity at each stage.

Reported in Tan & Le (2021), a comparison of the EfficientNet V2-S, which contains only 22M parameters (48% fewer than the EfficientNet-B6), with the EfficientNet-B6 reveals that the former achieves a comparable 83.9% top-1 ImageNet accuracy while undergoing up to four times faster training. This substantial enhancement in efficiency signifies the efficacy of EfficientNet V2’s architectural enhancements and training methodology in effectively surmounting the computational impediments that constrained the practical implementation of the original EfficientNet models. Since our images are larger than the EfficientNet’s largest input size, we also employed EfficientNet V2 to obtain higher accuracy and lower computational time.

Evaluation metrics

We first compute, four fundamental metrics: true positives (TP) and true negatives (TN) represent correctly classified positive and negative instances, respectively, whereas false positives (FP) and false negatives (FN) denote misclassifications where negative instances are erroneously identified as positive and positive instances as negative, respectively. The following well-known metrics are based on these fundamental metrics. Notably, in our multi-class classification tasks, we compute these metrics using a one-vs-all approach, where each class is evaluated against all other classes combined. The final metrics are obtained by averaging across all classes, weighted by their support (number of instances). Moreover, our use of five-fold cross-validation ensures the validity and feasibility of the employed models.

Accuracy: It represents the proportion of correct predictions among all predictions. While accuracy is intuitive, it may be misleading in imbalanced datasets where one class significantly outnumbers the other. Thus, more metrics are required for a fair evaluation.

Accuracy=TP+TNTP+TN+FP+FN.

Precision: Precision, or positive predictive value, measures the proportion of correct positive predictions among all positive predictions. This metric is particularly important when false positives have significant consequences, such as unnecessary medical procedures or treatments.

Precision=TPTP+FP.

Recall: Recall, also known as sensitivity or true positive rate, measures the proportion of actual positive cases that were correctly identified.

Recall=TPTP+FN.

Specificity: Specificity, or true negative rate, measures the proportion of actual negative cases that were correctly identified. In medical diagnosis, specificity is particularly crucial as it represents the prediction model’s ability to identify healthy individuals correctly.

Specificity=TNTN+FP.

F1-score: The F1-score is the harmonic mean between precision and recall, offering a balanced assessment of a model’s performance. It’s particularly valuable when equilibrium between these two metrics is needed, as it gives lower scores when either precision or recall is significantly imbalanced.

F1-score=2×Precision×RecallPrecision+Recall=2TP2TP+FP+FN

Area under the receiver operating characteristic (ROC) curve (AUC): The AUC represents the model’s ability to discriminate between classes across all possible classification thresholds. The AUC-ROC measures model performance by calculating the area beneath the ROC curve, which displays the true positive rate (recall) plotted against the false positive rate (1—specificity). A perfect classifier achieves an AUC of 1.0, while random guessing produces an AUC of 0.5.

Experimental design

In the experiment phase shown in Fig. 2, we aimed to build several predictive models for different types of EMA-eq classifications targeting various classes such as 2 (binary: negative/positive) and 3/4 (multi-class: negative, weak positive, strong positive, gray zone). Since we have a limited number of samples, to assess the performance of models fairly, we applied fivefold cross-validation coupled with stratified sampling, which ensures that a constant proportion of class samples are included in the training and validation portions of each fold.

Figure 2 An overview of our entire workflow depicting stages for (i) data splitting, (ii) data augmentation, and (iii) deep learning model training/validation/reporting.

To build and train various EfficientNet and EfficientNet-V2 models we employed Pytorch 2.5 framework along with several libraries such as (a) Albumentations for augmenting input images, (b) Sklearn for stratified k-fold sampling and metric computations, (c) Python Imaging Library (PIL) for image loading and (d) Wandb for performance tracking.

During the training phase, input images of approximately 1,170 × 980 pixels wide are first resized to the appropriate resolution (e.g., 224 × 224 for EfficientNet-B0) according to the specific input size of each architecture. Moreover, to increase the generalization ability and maintain robustness to new samples, we applied several augmentation techniques. Image augmentation techniques artificially expand training datasets by applying various transformations to existing images. Horizontal and vertical flips create mirror images along their respective axes, helping models learn orientation-invariant features. Rotation and scaling operations change the angle and size of objects within images, improving robustness to different viewpoints and object scales. Elastic transformations apply non-linear deformations that simulate natural variations in shape and structure, particularly useful for medical imaging and handwriting recognition. Grid distortion introduces controlled geometric distortions by warping images through a deformed grid pattern, creating realistic variations that help models generalize better to real-world scenarios where objects may appear stretched, compressed, or slightly warped due to camera angles or lens distortion. The details of applied augmentation are presented with their parameters in Table 2. The selection of given augmentation techniques is sourced from the inherent visual structures of the images we encounter.

Table 2 The applied augmentations’ parameters.

Augmentation type	Probability	Parameters (Key/Value)	
HorizontalFlip	0.4	N/A	
VerticalFlip	0.4	N/A	
ShiftScaleRotate	0.3	shift-limit: 0.1, scale-limit: 0.15, rotate-limit: 0.15	
ElasticTransform	0.05	alpha: 1, sigma: 50	
GridDistortion	0.05	num-steps: 5, distort-limit: 0.3	

During the training phase, we examined eight EfficientNet (i.e., B0–B7) and three EfficientNet-V2 (small, medium, large) architectures. We used the following EfficientNet architectures with resized square images ranging from B0: 224 × 224, B1: 240 × 240, B2: 260 × 260, B3: 300 × 300, B4: 380 × 380, B5: 456 × 456, B6: 528 × 528 and B7: 600 × 600 pixels whereas EfficientNet-V2 models were used with 640 × 640 pixels wide images. We hypothesize that larger images fed into the larger models will give better performance.

In Table 3, we introduce our key training properties by also involving the hyper-parameters we chose. We used label smoothing since it improves model generalization by preventing overconfidence in predictions, converting hard one-hot encoded targets into soft probability distributions that discourage the model from becoming too certain about its training examples and thus reducing overfitting, particularly when dealing with noisy or potentially incorrect labels in the dataset. In addition, we applied five-fold stratified cross-validation to provide more reliable performance estimates than random splitting, while making efficient use of the entire dataset by training on 80% and validating on 20% in each iteration. Ultimately, this reduces the variance in performance metrics and allows for better model selection. Followingly, we leveraged Weights & Biases (Wandb) tool to meticulously track our experiments, enabling real-time monitoring of performance metrics across multiple model configurations.

Table 3 The key properties and values used in the training phase.

Parameter	Value	
Optimizer	AdamW	
Optimizer learning rate	0.05	
Optimizer weight decay	0.03	
Loss criterion	Cross-entropy loss	
Label smoothing	0.2	
Number of epochs	30	
Number of K-folds	5	
Batch size	4	

Results

In this section, we analyze and interpret the results for three different dataset configurations of IgA EMA-eq Binary (negative and positive), IgA EMA-eq 3-Classes (negative, weak positive and strong positive) and IgA EMA-eq 4-Classes (negative, weak positive, strong positive and gray-zone) from both medical and machine learning points of view.

IgA EMA-eq binary: “detection of disease”

The performance metrics for all models, as shown in Table 4, demonstrate the high efficacy of deep learning approaches in automating EMA-eq test interpretation. The EfficientNetV2-S model achieved the highest overall performance with 99.37% ( ±0.7%) accuracy, 99.39% ( ±0.7%) precision, 99.37% ( ±0.7%) recall, and 99.36% ( ±0.7%) specificity. This model also demonstrated excellent discriminative ability with an AUC of 99.70% ( ±0.3%). These results indicate that the EfficientNetV2-S architecture can reliably distinguish between positive and negative EMA-eq samples with near-perfect accuracy.

Table 4 5-fold cross-validation results for IgA EMA-eq binary classification (i.e., positive/negative) using various EfficientNet architectures.

The results, indicated by parentheses, denote the mean and standard deviation scores, respectively.

Model	Accuracy	Precision	Recall	Specificity	F1-score	AUC	
Efficient Net B0	96.86% ( ±1.9%)	96.98% ( ±1.9%)	96.86% ( ±1.9%)	96.97% ( ±1.9%)	96.86% ( ±1.9%)	98.71% ( ±1.0%)	
Efficient Net B1	96.54% ( ±2.5%)	96.70% ( ±2.3%)	96.54% ( ±2.5%)	96.62% ( ±2.4%)	96.54% ( ±2.5%)	98.80% ( ±1.1%)	
Efficient Net B2	97.48% ( ±2.5%)	97.57% ( ±2.5%)	97.48% ( ±2.5%)	97.51% ( ±2.5%)	97.48% ( ±2.5%)	98.30% ( ±1.2%)	
Efficient Net B3	96.86% ( ±2.2%)	96.91% ( ±2.2%)	96.86% ( ±2.2%)	96.92% ( ±2.2%)	96.86% ( ±2.2%)	99.03% ( ±0.8%)	
Efficient Net B4	98.11% ( ±1.8%)	98.17% ( ±1.8%)	98.11% ( ±1.8%)	98.13% ( ±1.8%)	98.11% ( ±1.8%)	99.58% ( ±0.6%)	
Efficient Net B5	98.43% ( ±1.4%)	98.51% ( ±1.3%)	98.43% ( ±1.4%)	98.43% ( ±1.4%)	98.43% ( ±1.4%)	99.58% ( ±0.4%)	
Efficient Net B6	97.80% ( ±2.1%)	97.90% ( ±2.0%)	97.80% ( ±2.1%)	97.84% ( ±2.1%)	97.80% ( ±2.1%)	99.50% ( ±0.6%)	
Efficient Net B7	97.17% ( ±2.3%)	97.35% ( ±2.1%)	97.17% ( ±2.3%)	97.30% ( ±2.2%)	97.17% ( ±2.3%)	99.40% ( ±0.6%)	
Efficient Netv2 S	99.37% ( ±0.7%)	99.39% ( ±0.7%)	99.37% ( ±0.7%)	99.36% ( ±0.7%)	99.37% ( ±0.7%)	99.70% ( ±0.3%)	
Efficient Netv2 M	99.06% ( ±0.7%)	99.09% ( ±0.7%)	99.06% ( ±0.7%)	99.03% ( ±0.7%)	99.06% ( ±0.7%)	99.34% ( ±0.8%)	
Efficient Netv2 L	99.06% ( ±1.2%)	99.11% ( ±1.1%)	99.06% ( ±1.2%)	99.07% ( ±1.2%)	99.06% ( ±1.2%)	99.66% ( ±0.4%)	
Note:

The best scores are highlighted in bold.

Interestingly, among the traditional EfficientNet models (B0–B7), the EfficientNet-B5 performed best with 98.43% ( ±1.4%) accuracy, slightly outperforming both smaller (B0–B4) and larger (B6-B7) variants. This finding challenges the assumption that larger models consistently deliver better performance, suggesting that the B5 architecture provides an optimal balance between model complexity and generalization ability for this specific task.

The confusion matrices presented in Fig. 3 provide further insight into the classification patterns. Even the worst-performing model (EfficientNet-B0) demonstrated relatively strong performance, correctly classifying 27 negative and 32 positive samples in the displayed fold, with only two false positives and two false negatives. The average model (EfficientNet-B4) showed improved performance with 30 correctly classified negative and 33 positive samples, with only 1 misclassification in the positive class. The best model (EfficientNetV2-S) demonstrated near-perfect classification with 30 correct negative and 34 correct positive classifications, with only minimal misclassification (exists in other folds).

Figure 3 Confusion matrices selected from the (A) worst, (B) average and (C) best model in our binary classification experiments.

Our analysis reveals a general trend where increasing input resolution correlates with improved classification accuracy across the EfficientNet family, with models B0 through B5 showing consistent performance gains as resolution increases (from 96.86% to 98.43%). However, this pattern doesn’t hold universally, as evidenced by the slight performance decline in larger architectures like EfficientNet-B6 (97.80%) and B7 (97.17%), suggesting that extremely high resolutions may introduce diminishing returns or even slight overfitting for this specific IgA EMA-eq classification task.

Overall, all tested architectures achieved high performance metrics, with accuracy scores ranging from 96.54% to 99.37%, indicating the robustness of the EfficientNet family for this application. The slightly lower standard deviations observed in EfficientNetV2 models ( ±0.7% for S and M variants) compared to traditional EfficientNet architectures suggest greater consistency across different folds of the dataset. These results demonstrate that the EfficientNetV2 series has great potential for recognizing IgA-EMA tests for diagnosing celiac disease when fed a large-scale dataset. This could potentially reduce reliance on labor-intensive and subjective manual evaluation methods.

IgA EMA-eq 3-classes: “separating positiveness”

In this subsection, we present the results of our three-class classification experiments for IgA EMA-eq tests, where samples were categorized as negative, weak positive, or strong positive. Table 5 provides a comprehensive overview of the performance metrics across various EfficientNet architectures.

Table 5 Five-fold cross-validation results for IgA EMA-eq three-classes classification (i.e., strong positive/weak positive/negative) using various EfficientNet architectures.

The results, indicated by parentheses, denote the mean and standard deviation scores, respectively.

Model	Accuracy	Precision	Recall	Specificity	F1-score	AUC	
Efficient Net B0	91.84% ( ±3.0%)	92.23% ( ±3.0%)	91.84% ( ±3.0%)	95.69% ( ±1.6%)	91.77% ( ±3.0%)	97.53% ( ±0.5%)	
Efficient Net B1	92.45% ( ±1.8%)	92.79% ( ±1.4%)	92.45% ( ±1.8%)	96.29% ( ±0.7%)	92.43% ( ±1.7%)	97.09% ( ±1.5%)	
Efficient Net B2	92.77% ( ±1.5%)	93.32% ( ±1.6%)	92.77% ( ±1.5%)	96.39% ( ±0.7%)	92.77% ( ±1.5%)	97.78% ( ±0.8%)	
Efficient Net B3	92.46% ( ±1.8%)	93.27% ( ±1.7%)	92.46% ( ±1.8%)	96.24% ( ±0.8%)	92.42% ( ±1.8%)	96.92% ( ±1.2%)	
Efficient Net B4	94.98% ( ±1.8%)	95.37% ( ±1.5%)	94.98% ( ±1.8%)	97.50% ( ±0.8%)	94.99% ( ±1.7%)	98.57% ( ±0.4%)	
Efficient Net B5	94.02% ( ±1.1%)	94.43% ( ±1.2%)	94.02% ( ±1.1%)	97.13% ( ±0.6%)	94.04% ( ±1.1%)	98.46% ( ±0.5%)	
Efficient Net B6	94.02% ( ±1.1%)	94.17% ( ±1.2%)	94.02% ( ±1.1%)	97.06% ( ±0.5%)	93.98% ( ±1.2%)	98.19% ( ±0.5%)	
Efficient Net B7	94.02% ( ±1.8%)	94.45% ( ±1.9%)	94.02% ( ±1.8%)	96.85% ( ±0.9%)	93.92% ( ±1.9%)	98.24% ( ±1.6%)	
Efficient Netv2 S	95.28% ( ±3.1%)	95.84% ( ±2.6%)	95.28% ( ±3.1%)	97.68% ( ±1.4%)	95.23% ( ±3.1%)	98.42% ( ±1.1%)	
Efficient Netv2 M	94.97% ( ±2.2%)	95.35% ( ±2.1%)	94.97% ( ±2.2%)	97.38% ( ±1.1%)	94.88% ( ±2.3%)	98.41% ( ±0.9%)	
Efficient Netv2 L	94.66% ( ±3.0%)	94.74% ( ±3.1%)	94.66% ( ±3.0%)	97.30% ( ±1.5%)	94.57% ( ±3.1%)	98.02% ( ±1.7%)	
Note:

The best scores are highlighted in bold.

The EfficientNetV2-S model again demonstrated superior performance with the highest accuracy of 95.28% ( ±3.1%), precision of 95.84% ( ±2.6%), and recall of 95.28% ( ±3.1%). This model also achieved an impressive specificity of 97.68% ( ±1.4%) and F1-score of 95.23% ( ±3.1%), with an AUC of 98.42% ( ±1.1%). These results indicate that the EfficientNetV2-S architecture can effectively distinguish between the three different EMA-eq classes with high reliability.

Among the traditional EfficientNet models (B0–B7), we observed that EfficientNet-B4 delivered the best performance with 94.98% ( ±1.8%) accuracy, 95.37% ( ±1.5%) precision, and 94.98% ( ±1.8%) recall. Similar to our binary classification findings, there was not a linear improvement in performance with increasing model size, as EfficientNet-B4 outperformed larger variants like B5, B6, and B7, suggesting an optimal architecture size for this specific classification task. The confusion matrices presented in Fig. 4 provide deeper insights into the classification patterns. The worst-performing model (EfficientNet-B0) showed reasonable discrimination ability but struggled with distinguishing between weak positive and strong positive samples. The matrix reveals 27 correctly classified negative samples, but shows several misclassifications between weak positive and strong positive categories. The average model (EfficientNet-B3) demonstrated improved class separation with better classification accuracy across all three classes. The best model (EfficientNetV2-S) showed the clearest class separation with 30 correctly classified negative samples and substantially improved discrimination between weak positive and strong positive categories.

Figure 4 Confusion matrices selected from the (A) worst, (B) average and (C) best model in our three-class classification experiments.

A notable pattern across all models is the relative difficulty in distinguishing between weak positive and strong positive classes compared to identifying negative samples. This is consistent with the clinical reality where differentiation between varying degrees of positive results can be more challenging than distinguishing between positive and negative outcomes.

The superior performance of EfficientNetV2 models in the three-class classification task can be attributed to several key factors. First, the significantly higher input resolution of 640 × 640 pixels (compared to 224 × 600 pixels in standard EfficientNet variants) allows these models to capture more fine-grained textural and pattern details in the immunofluorescence images, which is particularly critical for distinguishing between subtle differences in weak positive and strong positive samples. Additionally, EfficientNetV2’s architectural innovations, including the Fused-MBConv blocks in early layers and non-uniform scaling strategy, appear particularly well-suited for handling the increased complexity of multi-class discrimination tasks. The model’s adaptive regularization technique likely contributes to its improved generalization capabilities across the three classes while its more efficient training dynamics may allow it to reach a better optimization point within the same number of training epochs. This performance advantage highlights that for nuanced medical image classification tasks like EMA-eq pattern recognition, the combination of higher resolution inputs and EfficientNetV2’s architectural improvements provides a substantial benefit in discriminative power.

Overall, the multi-class classification performance demonstrates that EfficientNet architectures, particularly EfficientNetV2-S, can effectively automate the more nuanced three-class categorization of IgA EMA-eq tests with high accuracy, potentially enhancing the diagnostic precision for celiac disease beyond simple binary classification.

IgA EMA-eq 4-classes: “where the ambiguity comes in”

Analysis of the four-class classification results (negative, weak positive, strong positive, and gray-zone) reveals distinct performance patterns across the EfficientNet architectures, as demonstrated in Table 6 and Fig. 5. The four-class paradigm represents the most granular classification scenario in our study, introducing an additional gray-zone class that requires even finer discrimination capabilities.

Table 6 5-fold cross-validation results for IgA EMA-eq four-classes classification (i.e., strong positive/weak positive/negative/gray zone) using various EfficientNet architectures.

The results, indicated by parentheses, denote the mean and standard deviation scores, respectively.

Model	Accuracy	Precision	Recall	Specificity	F1-score	AUC	
Efficient Net B0	81.79% ( ±2.9%)	83.12% ( ±2.8%)	81.79% ( ±2.9%)	93.79% ( ±1.0%)	81.19% ( ±2.8%)	92.69% ( ±1.5%)	
Efficient Net B1	81.80% ( ±2.4%)	82.54% ( ±2.5%)	81.80% ( ±2.4%)	93.98% ( ±0.7%)	81.90% ( ±2.3%)	92.97% ( ±2.6%)	
Efficient Net B2	85.34% ( ±3.9%)	86.12% ( ±4.0%)	85.34% ( ±3.9%)	95.12% ( ±1.2%)	85.29% ( ±4.2%)	93.78% ( ±2.3%)	
Efficient Net B3	82.34% ( ±1.4%)	84.53% ( ±1.6%)	82.34% ( ±1.4%)	94.16% ( ±0.4%)	82.48% ( ±1.5%)	93.12% ( ±1.0%)	
Efficient Net B4	83.45% ( ±6.0%)	85.87% ( ±3.7%)	83.45% ( ±6.0%)	94.71% ( ±1.8%)	83.98% ( ±5.6%)	94.18% ( ±3.2%)	
Efficient Net B5	86.15% ( ±2.1%)	86.93% ( ±1.8%)	86.15% ( ±2.1%)	95.51% ( ±0.7%)	86.21% ( ±2.1%)	94.30% ( ±2.9%)	
Efficient Net B6	82.35% ( ±4.8%)	84.05% ( ±3.9%)	82.35% ( ±4.8%)	94.43% ( ±1.4%)	82.75% ( ±4.3%)	92.80% ( ±4.4%)	
Efficient Net B7	83.72% ( ±4.7%)	84.85% ( ±4.9%)	83.72% ( ±4.7%)	94.51% ( ±1.5%)	83.64% ( ±4.9%)	92.73% ( ±4.9%)	
Efficient Netv2 S	86.98% ( ±4.8%)	87.10% ( ±5.3%)	86.98% ( ±4.8%)	95.80% ( ±1.5%)	86.79% ( ±5.2%)	95.98% ( ±1.9%)	
Efficient Netv2 M	86.98% ( ±4.6%)	87.38% ( ±4.6%)	86.98% ( ±4.6%)	95.57% ( ±1.5%)	86.32% ( ±5.4%)	95.25% ( ±2.2%)	
Efficient Netv2 L	85.06% ( ±2.6%)	85.12% ( ±3.8%)	85.06% ( ±2.6%)	95.22% ( ±0.9%)	85.00% ( ±3.3%)	94.08% ( ±3.0%)	
Note:

The best scores are highlighted in bold.

Figure 5 Confusion matrices selected from the (A) worst, (B) average and (C) best model in our four-class classification experiments.

Once again, the EfficientNetV2-S model achieved the highest overall performance with 86.98% ( ±4.8%) accuracy, 87.10% ( ±5.3%) precision, and 86.98% ( ±4.8%) recall, with a high AUC of 95.98% ( ±1.9%). While the EfficientNetV2-M model showed identical accuracy, the S variant demonstrated a slightly better overall performance profile, particularly with its superior specificity, F1 and AUC score. This performance excellence can be attributed to the higher input resolution of 640 × 640 pixels used for EfficientNetV2 models, which provides crucial additional detail for discriminating between the closely related classes, particularly the complex gray-zone category.

Among the traditional EfficientNet architectures, EfficientNet-B5 demonstrated the best performance with 86.15% ( ±2.1%) accuracy, continuing the pattern observed in previous classification schemes where mid-to-large size models (rather than the largest variants) deliver optimal results. This finding reinforces the notion that an optimal balance between model complexity and generalization capability is more important than sheer model size for this specific medical imaging task.

The confusion matrices in Fig. 5 illustrate the classification challenges in this more complex four-class scenario. The worst-performing model (EfficientNet-B0) shows particular difficulty in correctly identifying gray-zone samples and distinguishing between weak positive and strong positive categories, as evidenced by the multiple off-diagonal entries. The average model (EfficientNet-B2) demonstrates improved classification accuracy, particularly for negative samples, but still exhibits confusion between the positive classes and the gray zone. The best model (EfficientNetV2-S) shows the most balanced performance across all four classes, with notably improved classification of gray-zone samples compared to other architectures. The overall reduction in accuracy metrics compared to binary and three-class classification schemes (dropping from 99% to 87% for the best models) highlights the inherent difficulty in distinguishing between four clinically relevant categories, particularly the gray zone class, which represents borderline cases. This performance decrease is expected given the increased classification complexity and the inherent ambiguity of gray-zone samples, which represent complex cases even for human experts. The consistently strong performance of EfficientNetV2 models across all classification schemes, particularly with higher input resolutions, suggests that these advanced architectures, combined with detailed image inputs, provide the most reliable automated approach for IgA EMA-eq test interpretation, even in the most granular multi-class scenarios.

Evaluation on initially titrated samples

In this part, we evaluate our most successful trained models for each category on a limited but initially titrated samples. This external dataset contains unique (i.e., never used in previous stages) 30 samples covering four classes (i.e., Neg: 5, GrayZone: 5, Weak-Positive: 10, Strong-Positive: 10). Table S1 introduces actual images and details of these samples, whereas Table 7 demonstrates our best ML models’ performance for each mode on this external dataset. Note that 2-Class mode unites all positive samples into one single positive class while four-Class mode has all distinct classes. Thus, this evaluation protocol uses the same structure presented in the previous three subsections. As given in Table 7, our selected models achieved an accuracy of 100% in binary and triple class scenarios, surpassing the average scores in related five-fold cross-validation results. For the four-class scenario, our best model achieved an accuracy of 83.33% and a specificity score of 95%. Although the grayzoned test case shows a slightly lower result than our average accuracy score of 86.98%, the benchmarking yields error-free predictions in binary and triple class regimes.

Table 7 Performance measurement of the three best models across each classification category using the data of initially titrated samples.

Mode	Model	Accuracy	Precision (macro)	Recall (macro)	Specificity (macro)	F1-score (macro)	AUC	
2-Class	EfficientNetV2-S	100%	100%	100%	100%	100%	100%	
3-Class	EfficientNetV2-S	100%	100%	100%	100%	100%	100%	
4-Class	EfficientNetV2-S	83.33%	87.50%	85.00%	95.00%	82.64%	96.25%	

Comparative study

To evaluate the comparative advantage of our approach with existing methods, a comparative study was conducted. In this regard, we selected the algorithm (local binary patterns + support vector machine: LBP+SVM) which was used in the work of Caetano dos Santos et al. (2019). Although they used human umbilical cord (HUC), which is different than ours (monkey liver—ML), the visual similarity of images and the domain similarity of the work motivated us to compare the LBP+SVM method with EfficientNet-V2. We, thus, reproduced the implementation of Caetano dos Santos et al. (2019), which uses LBP-based feature extraction and the AdaBoost-powered SVM algorithm. Meanwhile, this benchmark also directly compares the use of hand-crafted features with the representation learning offered by our scheme. To ensure a fair comparison, we used a five-fold cross-validation that is identical to ours. We then compared the results of both methods, in particular with our best binary, 3-class and 4-class models. The results are given in Table 8.

Table 8 Comparative results obtained with other method(s).

Study	Substrate	Mode	Algorithm	Avg. accuracy	Avg. sensitivity	Avg. specificity	
Caetano dos Santos et al. (2019)	HUC	2-Class	LBP + SVM	87.14%	85.80%	88.59%	
Ours	ML	2-Class	EfficientNetV2-S	99.37%	99.37%	99.26%	
Caetano dos Santos et al. (2019)	HUC	3-Class	LBP + SVM	74.52%	74.52%	87.02%	
Ours	ML	3-Class	EfficientNetV2-S	95.28%	95.28%	97.68%	
Caetano dos Santos et al. (2019)	HUC	4-Class	LBP + SVM	59.52%	59.52%	85.86%	
Ours	ML	4-Class	EfficientNetV2-S	86.98%	86.98%	95.80%	
Note:

HUC, Human umbilical cord; ML, Monkey liver

Our work significantly advances beyond (Caetano dos Santos et al., 2019) by transitioning from traditional handcrafted feature extraction methods to state-of-the-art deep learning architectures. While they employed LBP, a conventional computer vision technique that relies on manually designed texture descriptors, our approach leverages EfficientNet and EfficientNetV2 models that automatically learn hierarchical feature representations directly from raw immunofluorescence images. This fundamental methodological shift from handcrafted to learned features enables our models to capture complex, subtle patterns in EMA images that traditional texture descriptors may miss. The superior performance achieved in all datasets compared to their LBP-based approach demonstrates the effectiveness of modern deep learning architectures in medical image analysis, particularly for tasks requiring fine-grained pattern recognition in immunofluorescence microscopy.

Discussion

In this section, we discuss the results from both the machine learning and clinical laboratory perspectives. In addition, we have examined the built models through the lens of HiRes-CAM, a CNN compatible and explanatory AI companion module that shows heat maps of the pixel regions from which the model significantly derives its results. In this context, we evaluate the decisions of our best model (i.e., EfficientNetV2-S) on correct and incorrect predictions to (a) better understand the behavior of the model and (b) what could be done for better performance in terms of data and model improvements.

Machine learning perspective

Our comprehensive evaluation of EfficientNet architectures across binary, 3-class, and 4-class IgA EMA-eq classification schemes reveals several significant patterns and insights.

First, we observed a consistent decrease in classification performance as task complexity increased from high accuracy (99.37%) in binary classification to strong performance (95.28%) in three-class classification, and finally to good performance (86.98%) in the most complex four-class scenario. This pattern reflects the inherently increasing difficulty in distinguishing between more granular diagnostic categories, particularly when dealing with subjective boundaries between classes like weak positive, strong positive, and gray-zone samples.

Second, EfficientNetV2 models, particularly the EfficientNetV2-S variant, demonstrated superior performance across all classification tasks. This excellence can be attributed to both architectural innovations (fused-MBConv blocks, non-uniform scaling) and the higher input resolution (640 × 640 pixels) used with these models, which appears particularly beneficial for capturing the subtle immunofluorescence patterns critical for accurate classification. The consistently high AUC values (99.70%, 98.42%, and 95.98% for binary, three-class, and four-class tasks, respectively) demonstrate the robust discriminative ability of these models. It is also noteworthy that the high predictive performance of these models is also related to the visual augmentations used during training. We observed that the absence of all augmentations resulted in a drop in performance of around 4.5%. Thus, deep learning practitioners should carefully consider the augmentations under the hood when they are short of available labeled data.

Third, contrary to what might be expected, the largest models did not necessarily deliver the best performance. Among traditional EfficientNet architectures, mid-to-large variants (B4/B5) consistently outperformed both smaller (B0–B3) and larger models (B6–B7). This finding suggests that an optimal balance between model complexity and generalization capability is more critical than sheer model size for this specific medical imaging application.

Despite these promising results, our study has limitations that warrant further investigation. First, the relatively small dataset size (254 samples for binary and three-class, 294 for four-class classification) may limit the models’ generalizability to diverse clinical settings. Though we have used five-fold cross validation during experimentation, future work should focus on expanding the dataset to include samples from multiple institutions and diverse patient populations. Second, while our models perform well on high-quality images, their robustness to variations in immunofluorescence image quality, which commonly occur in routine clinical practice, remains to be evaluated.

Additionally, the confusion matrices and HiRes-CAM visualizations reveal persisting challenges in distinguishing between closely related classes, particularly between gray-zone and other categories. This suggests that future research could benefit from exploring ensemble approaches that combine multiple model predictions or integrating additional clinical data to improve discrimination between borderline cases.

The results demonstrate that deep learning approaches, particularly using EfficientNetV2 architectures with higher resolution inputs, can effectively automate IgA EMA-eq test interpretation with high accuracy across different classification granularities. This automation potential could significantly reduce the subjectivity and labor-intensive nature of current manual evaluation methods, ultimately improving celiac disease diagnostic workflows in clinical settings.

Clinical laboratory perspective

From a laboratory medicine standpoint, this study presents a practical and clinically relevant approach to standardizing the interpretation of EMA-eq tests. In a previous publication, Caetano dos Santos et al. (2019) presented one of the first studies to implement machine learning in the problem domain and their findings underscore that machine learning-based approaches can substantially reduce subjectivity in EMA interpretation, paving the way for more consistent and scalable celiac disease diagnostics. Unlike conventional workflows that rely on serial dilutions for titer estimation, our methodology involves direct matching of patient immunofluorescence images to a reference panel composed of consensus-rated examples, each representing a discrete staining category (ranging from negative to strong positive). This streamlined strategy not only reduces processing time and laboratory workload but also minimizes inter-observer variability by providing fixed visual anchors for classification (Boral & Togay, 2023).

To ensure clinical validity, we excluded patients with selective IgA deficiency, as their inclusion could result in falsely negative EMA-eq interpretations and compromise both model training and the reliability of test outputs. A structured pre-assessment phase was implemented to further enhance standardization. Before the main evaluation, expert raters reviewed a curated set of reference images for each category. This step was crucial for harmonizing classification criteria across observers, particularly in complex cases such as weak positives and gray-zone samples. As shown in Table 7, performance metrics on the initially titrated samples’ images remained high under both the 2-Class and 3-Class modes, demonstrating strong alignment with dilution-based ground truth labels. The EMA-eq 4-class configuration introduced the gray zone as an additional interpretive category, which posed significant challenges for both human and machine-based evaluation. These gray zone cases typically exhibit faint or heterogeneous staining patterns on monkey liver tissue, lacking the clear-cut fluorescence observed in higher-titer samples. As a result, they may be misclassified by convolutional neural networks, and even experienced evaluators may struggle to confidently assign them to either the negative or positive class. This ambiguity is further illustrated by the HiRes-CAM visualizations presented in Figs. 6–9. In Fig. 6, the gray zone examples correctly predicted by EfficientNetV2-S (e.g., Figs. 6A, 6B) underscore the model’s strength in handling borderline cases. Notably, in Fig. 6D, the model incorrectly classifies a gray zone image as weakly positive. However, this misclassification aligns with the experience of expert raters, who also reported uncertainty and reached consensus only after considering that the patient’s anti-tTG IgA level was negative—ultimately supporting a gray zone interpretation. Such cases highlight both the challenges of gray zone classification and the potential of AI to reflect nuanced human-like judgments. In Fig. 7, we further examine ground-truth negative images. We deliberately selected samples exhibiting non-specific background staining to evaluate model robustness. Despite these visual confounders, the model correctly identified clearly negative cases (Figs. 7A, 7B). For more ambiguous samples, such as Fig. 7D, the model predicted a weakly positive label. Yet this particular patient also had a negative anti-TTG IgA result and was clinically asymptomatic, prompting us to categorize the prediction as a false positive. Nonetheless, such borderline images are likely to elicit confusion even among experienced human readers, reaffirming the difficulty of these cases.

Figure 6 A sample HiRes-CAM visualization of our best four-class model, EfficientNet-V2-S, on actual grayzone images.

Each image shows both ground truth and predicted labels along with the confidence score.

Figure 7 A sample HiRes-CAM visualization of our best four-class model, EfficientNet-V2-S, on actual negative images.

Each image shows both ground truth and predicted labels along with the confidence score.

Figure 8 A sample HiRes-CAM visualization of our best four-class model, EfficientNet-V2-S, on actual weak positive images.

Each image shows both ground truth and predicted labels along with the confidence score.

Figure 9 A sample HiRes-CAM visualization of our best four-class model, EfficientNet-V2-S, on actual strong positive images.

Each image shows both ground truth and predicted labels along with the confidence score.

Figure 8 provides further support for the utility of the four-class approach. Although some false predictions are observed, many of these are arguably acceptable when viewed from a clinical context. For instance, some images predicted as gray zone could reasonably be interpreted as such by expert raters. While our binary model excluded gray zone samples due to their intrinsic ambiguity, our findings suggest that this four-class model—particularly when used for screening—may offer substantial practical advantages in laboratory workflows by maintaining high sensitivity while reducing ambiguity. Additionally, the HiRes-CAM outputs in Fig. 9 highlight the model’s reliability in recognizing high positive images. However, in Fig. 9D, the model classified a strongly positive EMA-eq image as weakly positive, likely influenced by nuclear homogeneous staining patterns superimposed on liver tissue—possibly related to ANA cross-reactivity. This observation suggests a need to investigate further the impact of ANA positivity on EMA-eq test interpretation in future studies.

Importantly, inter-observer disagreement is a known characteristic of EMA-eq interpretation, particularly in low-intensity or borderline cases. While segmenting positivity into multiple subclasses may increase subjectivity and reduce reproducibility, it can offer meaningful granularity for clinical monitoring (Anbardar et al., 2023). In this context, the added detail may support nuanced decision-making in treatment follow-up, despite the inherent challenges in achieving perfect agreement. Moreover, in real-world settings, certain EMA-eq images may contain damaged cellular structures or partially void regions due to technical limitations in slide preparation or imaging (Hocke et al., 2023). These imperfections pose interpretation challenges not only for automated systems but also for human experts. As such, attributing misclassification solely to artificial intelligence may be misleading; rather, these issues reflect fundamental limitations of the indirect immunofluorescence technique itself (Meroni et al., 2019).

While the system has shown high agreement with expert interpretations, a key limitation is that its classification is entirely based on image-derived patterns and does not incorporate contextual serological data such as anti-tTG IgA levels. This restricts the model to a mimicry of human visual evaluation, rather than enabling it to contribute independent diagnostic value. Furthermore, no combined decision pipeline was implemented in which AI-generated predictions are revised or confirmed by human experts using additional clinical inputs. This absence limits its real-world applicability in borderline cases where interpretation requires more than fluorescence intensity alone. In addition, although the commercial assay used in this study is labeled as an EMA test, it employs monkey liver sections that anatomically lack true endomysial tissue. Therefore, the fluorescence patterns observed are more likely associated with reticulin antibody binding in the perisinusoidal framework. While these patterns are clinically accepted surrogates for EMA positivity—particularly in Euroimmun’s platform and when correlated with anti-tTG IgA results—this distinction may affect generalizability to other test systems that utilize classical EMA substrates, such as monkey esophagus or human umbilical cord. To enhance future applicability, cross-substrate validation across different platforms is recommended.

In clinical practice, gray zone findings often coincide with low anti-tTG IgA levels and serve as a diagnostic trigger for further investigations such as duodenal biopsy, as recommended by the ESPGHAN guidelines (Husby et al., 2020). Rather than reflecting simple uncertainty, the gray zone represents a diagnostic buffer space and transitional segment between seronegativity and early seropositivity. Importantly, this category may contain early or low-grade disease signals that warrant closer scrutiny (Piccialli et al., 2021). Although ESPGHAN guidelines recommend a sequential testing strategy for newly diagnosed coeliac disease, where EMA IIF is performed in patients with anti-TTG IgA levels ≥ 10 × ULN (upper limits of normal), this approach is not consistently followed in real-world clinical settings. In our institution, EMA-eq and anti-TTG IgA tests are often requested concurrently. This dual-request pattern is primarily driven by clinicians’ preference to maximize diagnostic sensitivity. Notably, of the 50 gray zone samples examined in our study, 14 were anti-TTG IgA positive. However, these levels did not exceed the 10 × ULN threshold. In the absence of duodenal biopsy data, we deliberately refrained from drawing categorical diagnostic conclusions based solely on serology. Nevertheless, clinicians may interpret weak positive or gray zone EMA results as possible indicators of early or evolving coeliac disease (Catassi & Fasano, 2008). This is especially true when the results are accompanied by even moderately elevated anti-TTG levels. This interpretive tendency further emphasizes the importance of refining model performance in borderline categories, where laboratory findings often exist in a diagnostic gray area and clinical decision-making proceeds under uncertainty. From a methodological perspective, these ambiguous cases reflect real-life diagnostic challenges and offer a valuable opportunity to test whether deep learning models can handle complex and uncertain situations—not just clear-cut positive or negative cases. Meanwhile, the current study involves other limitations. The relatively small sample size, particularly within the gray zone subgroup, may limit statistical power and generalizability. Most notably, the study does not include histopathological (biopsy) results, limiting the ability to stratify gray zone cases as true or false positives definitively.

Conclusions

Our comprehensive evaluation of EfficientNet and EfficientNet-V2 architectures for automated IgA EMA-eq test interpretation demonstrates the significant potential of deep learning approaches in celiac disease diagnosis. The results across binary, three-class, and four-class classification schemes consistently show that these models, particularly EfficientNetV2-S, achieve promising performance with accuracies of 99.37%, 95.28% and 86.98%, respectively. According to our best knowledge, this is the first study to apply these deep architectures to monkey liver–based EMA-eq immunofluorescence images using a structured reference panel and a dedicated gray-zone category, addressing a diagnostically challenging yet underexplored area. Several key findings emerge from our study. First, the performance degradation observed as classification granularity increases highlights the inherent difficulty in distinguishing between subtle immunofluorescence patterns, especially for gray-zone samples. This mirrors the challenges faced by human experts in clinical practice. Second, contrary to the conventional assumption that larger models yield better results, the model complexity must be carefully balanced against generalization capability for medical imaging tasks. Moreover, the problem domain exhibits challenges in discriminating closely related classes, particularly between gray-zone and other categories, underscoring the diagnostic ambiguity of such cases and the need for multimodal approaches. Integrating quantitative serological markers such as anti-tTG IgA levels or clinical metadata may help resolve these ambiguities and improve model robustness in borderline scenarios, especially when multimodal approaches are used. However, it is important to note that the present study utilized a monkey liver–based substrate, which lacks true endomysial tissue. While the observed fluorescence patterns are clinically interpreted as EMA positivity in commercial settings, they anatomically reflect reticulin fiber staining. As such, the direct generalizability of these findings to other substrates (e.g., monkey esophagus or human umbilical cord) should be approached with caution. Future studies should validate the proposed approach across multiple EMA testing platforms to ensure broader applicability. In conclusion, our findings demonstrate that deep learning approaches based on EfficientNet architectures can effectively automate reticulin based IgA EMA-eq test interpretation with high accuracy, potentially reducing reliance on subjective manual evaluation. By incorporating visually explainable outputs via HiRes-CAM and modeling real-world diagnostic uncertainty through a four-class configuration, our approach brings computer vision based serologic screening closer to practical clinical application. With further refinement and validation, these models could become valuable tools in clinical practice, enhancing the efficiency and reliability of celiac disease diagnosis while freeing specialized medical personnel for other critical tasks. Future work should focus on (i) prospective clinical validation, (ii) use of multimodal deep learning, (iii) improvements in model explainability, and (iv) the integration of these systems into existing diagnostic workflows to realize their full potential in enhancing patient care.

Supplemental Information

Supplemental Information 1 TTG Results.

We used Deepl Write and Claude for grammar checking and Claude for codebase debugging.

Additional Information and Declarations

Competing Interests

The authors declare that they have no competing interests.

Author Contributions

Mehmet Soylu conceived and designed the experiments, analyzed the data, authored or reviewed drafts of the article, and approved the final draft.

Ahmet Selman Bozkir conceived and designed the experiments, performed the experiments, analyzed the data, prepared figures and/or tables, authored or reviewed drafts of the article, and approved the final draft.

Human Ethics

The following information was supplied relating to ethical approvals (i.e., approving body and any reference numbers):

The Ethics Committee of Ege University Faculty of Medicine approved the study (Approval No: 2023-1750/23-11.2T/13).

Data Availability

The following information was supplied regarding data availability:

Data is available at Zenodo: Bozkir, A. S. (2025). Recognizing IgA-Class Endomysial Antibody Equivalent Binding Patterns on Monkey Liver Substrate Through EfficientNet Architectures and Deep Learning. In PeerJ. Zenodo. https://doi.org/10.5281/zenodo.17153082

The code is available at GitHub and Zenodo:

- https://github.com/asfix/iga-ema-classification

- Bozkir, A. S. (2025). Deep Learning for Celiac Diagnosis: Recognizing IgA-Class EMA Patterns on Monkey Liver Substrate Through EfficientNet Architectures. Zenodo. https://doi.org/10.5281/zenodo.16333476.

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
