# Peer review of "Recognizing IgA-class endomysial antibody equivalent binding patterns on monkey liver substrate through EfficientNet architectures and deep learning"

_PeerJ, doi:10.7717/peerj.20191_

## Round 0.1 · original submission · Major Revisions

Working on celiac disease, I believe this manuscript will be an essential contribution to the future of pediatric gastroenterology. I strongly suggest that the authors follow the reviewers' detailed requests. Please do not dismiss their contributions.

Reviewer 1 ·

Basic reporting

Introduction section:

The introduction section should provide more background information on the challenges and malpractice issues in diagnosing celiac disease. This should include statistics on misdiagnosis rates and the negative impacts that late diagnosis or malpractice can have on patients, the healthcare system, and the economy. Establishing this context would help readers better understand the significance and need for the proposed deep learning approach to improve diagnosis. The thorough information given between lines 34-52 raises questions about why the authors are providing these details, as they do not clearly explain the need for automated systems. Instead, the authors should focus on the main reasons for the difficulties in diagnosis and how deep learning can help address them.

Background/Related work section:

For this article, we can use keywords such as "Celiac diagnosis," "Deep Learning," "IGA EMA tests," and "image processing" to search relevant literature. Querying academic databases and Google Scholar with these keywords reveals numerous recent studies that the authors have not mentioned. Additionally, keywords like "Celiac diagnosis," "Deep Learning," "CNN," and "image processing" could uncover valuable research that is directly relevant to the topic. However, it is unclear why the authors include background information on skin cancer, COVID-19, and other unrelated datasets in lines 66-79, instead of focusing on the images obtained through invasive or non-invasive techniques for celiac disease diagnosis. The authors should focus on the application of CNNs and other deep learning techniques for diagnosing celiac disease from medical images, face images, and other relevant data to demonstrate the contribution and originality of their study. And It is crucial to highlight the gaps in the current literature that the authors aim to address with their research.

Materials and Methods Section:

Division of the “Data” section into two subsections such as data gathering and data preprocessing would improve the clarity and understanding of the methodology.

In the data gathering subsection, the authors should include the information provided between lines 165 and 170, as it would be better placed just before the demographic distribution details given in lines 114-119. The authors could include sample figure examples from the EMA datasets presented in Table 1 to provide readers with a better visual understanding of the data. Alternatively, they could refer to Table 5 to achieve a similar effect. Next, the authors could present the data preprocessing steps in more detail. As mentioned in lines 286-291, they should explain the preprocessing procedures they used, such as resizing, and augmentation. This could be accompanied by a brief figure to illustrate the full data gathering and preprocessing process, providing readers with a clearer understanding of how the raw data was transformed into the final input for the deep learning models.

Dividing the discussions section into "Machine Learning" and "Clinical Perspective" is an effective way to present this multidisciplinary study, as it would allow for a clearer communication of the different aspects and implications of the research findings. Separating limitations under a distinct section title would improve the clarity and readability for readers, allowing them to easily identify the gaps of the study.

In my view, the research design and proposed methodology in this publication are not entirely adequate. The research design would benefit from a substantial revision. Additionally, academic paraphrasing in English could enhance the overall understanding of the work. I recommend adding brief summary figures to illustrate the steps followed, in order to facilitate the reader's understanding.

Experimental design

I believe that improving the study design would be beneficial. The authors do not clearly articulate the research questions in their research design. The contribution and originality of using this method on this dataset are also not explicitly stated. As mentioned in the previous section, the background section does not clearly outline the existing literature on the use of different types of images for CD diagnosis, and how this study contributes to that literature.

Validity of the findings

I dont believe conclusions are well stated.

Additional comments

no comments.

Reviewer 2 ·

Basic reporting

The manuscript is well written.
I suggest transferring the description of the discrepant results illustrated by Figures 5-8 into the results section. Please provide a detailed table on grayzore results instead of narrative examples. Please add to this table transglutaminase antibody ELISA results as additional information.
How many were true low positives and unclears as to their patterns?

Figures 2-4: Please use larger, more legible fonts for the numbers in the confusion matrices.

Experimental design

The aim of the study was a practical and clinically relevant approach to standardizing the interpretation of endomysial antibody (EMA) tests. Since the recognition of the positive EMA pattern needs expertise and time, automation would help the broader application of this test in clinical practice. As an experimental design, the authors selected positive and negative test result source immunofluorescent images obtained in the test manufactured by Euroimmun, which uses monkey liver sections as the substrate.
Although this test is sold as an ’endomysial antibody’ assay, monkey liver does not contain muscle structures; thus, by definition cannot provide endomysial antibody results: the positivity should correctly be called reticulin antibody positivity. Although both endomysial and reticulin antibodies target the type-2 transglutaminase autoantigen and both are useful for the diagnosis of celiac disease, the visual patterns to be recognized as positive (the main topic of the developed solution) are substantially different from each other, based on different anatomical structures expressing transglutaminase. This fact interferes with the generalizability of the results and questions their utility for laboratories which use the classical endomysial antibody substrates, i.e. monkey esophagus or human umbilical cord, as shown by the low agreement with the evaluation based on the codes from the Caetano dos Santos-authored study, which were derived from human umbilical cord endomysial images.

Methods: The visual assignment of intensity may not be reliable to replace the antibody quantitation by titration in case of polyclonal antibodies, such as those in celiac disease. I suggest testing the machine-based evaluation also with the initially titrated 30 samples with the IgA-EMA 3-Classes strategy, and provide the results as a comparison to the whole dataset.

Validity of the findings

The authors provide diagnostically highly efficient results with their EfficientNet versions of image analysis for the recognition of clearly positive or negative samples, and this will likely be useful for helping the observer when faced with strange and unclear staining patterns.

Discuss and interpret the dissection of grayzone results into true low positives and false positives/unclears. Detecting true EMA positivity in low titer (1:10) is an important application field for the EMA testing and may be the basis of biopsy indication.

Given that the study was performed on liver sections, the results cannot be extrapolated to other commonly used EMA substrates (esophagus, umbilical cord). The conclusions/title should be modified to refer to the reticulin antibody positivity recognition.

Reviewer 3 ·

Basic reporting

The paper is mostly written in clear English. A mixture of British (e.g. labour) and American English (e.g. celiac) is used; it should be consistent.

Ideally, the paper would include more scientific and less opinionated language (e.g. exceptional performance, challenging four-class scenario)

The paper is significantly too long for the amount of material it covers. All sections apart from the introduction and related work section could be significantly shortened, which would help the reader.

Abstract: Explain the meaning of binary, three-class, four class in the second sentence.
Also explain what exactly is meant by positive negative etc. in the following sentence (e.g. does positive include both weakly and strongly positive etc.). Some of this is explained in the paper, but should be clear in the abstract as well.

The introduction is slightly repetitive (e.g. mentioning labour intensive process several times). Instead some readers would benefit from a more detailed introduction to EMA IgA.

There have been numerous studies developing AI algorithms for coeliac disease diagnosis in recent years, mostly on duodenal biopsies, but also on video capsules. and serological data. They should all be referenced.

How does this paper improve on Caetano dos Santos et al (2019)?

No need to focus so much on active learning, not that relevant to this paper

Experimental design

Line 114 says that there are a total of 294 samples, but then Table 1 includes 294 training samples and a further 74 validation samples. Please clarify these discrepancies and further detail how the data is split into training and validation samples. Later on you mention cross-validation, but the table is misleading.

Only performing cross-validation, would be good to have a test set, as otherwise you may overfit to the validation set.

Given that Efficientnet and Efficentnetv2 are published paper, the text describing the architectures can be shortened.

Specify how the microscope is fully automated. Also mention that 2 independent laboratory personal evaluated the cases. Where there any disagreements? If so how where those samples labelled?

Equivocal might be a more formal term than “grey zone”

Figure 2-4: include all 5 folds in each confusion matrix not just one randomly picked fold.

Figure 3: minor point, but positive-strong should not be in the middle
Figure 4: order from negative, gray zone (note it should be 2 words), positive-weak, positive-strong

Lines 342-344 I think are slightly misleading as the data is a toy dataset (only includes clear positives and negatives), so it doesn’t really indicate that AI can automate the diagnosis.

Don’t include words like “impressive” specificity in line 352.

Combine Figures 5-8 into one large Figure that fits in one page. Move the rest to the appendix.

Move the HiRes-CAM analysis to the Results section.

Validity of the findings

Data and code have been provided and results have been evaluated with a large number of significant metrics.

Additional comments

Limited real world applicability as ML is prone to overfitting on data from a single source. Would be beneficial to test on data from a different lab.

Would be good to get inter-observer agreement (using human experts) between cases.

Sometimes it’s referred to as “strongly positive” and sometimes as “positive strong”. Should be consistent; the first version is preferred.


The paper covers an interesting problem. The results are encouraging. And the comparison between different Efficientnet models is interesting.
However, the usefulness of the 2 and 3 class experiments are oversold, the paper could be shortened, and a proper tests set could be added (ideally from a different source, but even a test set from the same source would be beneficial.)

---

## Round 0.2 · Major Revisions

Although a reviewer recommended accepting this revision, another reviewer has raised several criticisms and comments. Please address them thoroughly.

Reviewer 2 ·

Basic reporting

1) The manuscript has been improved by the revision, but in its current form still does not answer adequately the main criticism and does not use professional terminology for the celiac antibodies.
Endomysial antibody positivity on liver (sic) is a contradiction of terms.
Even if used by a producer of a diagnostic kit for marketing, it is not acceptable in a scientific publication. Accordingly, at all instances mentioned, it should be amended and replaced to make a clear distinction from classic EMA, not just mentioned in the discussion as a possible limitation.(For your information, two other companies marketed their anti-tTG ELISA kits as endomysial tests, still no health care professional regards them as true EMA and the authors likely would agree with this.)

There may be two ways to fix this issue, either to use the designation "EMA equivalent" (e.g. EMA-eq in abbreviation) for all own results and their discussion, or to strictly restrict (and redo) the image analysis to the portal vessels (arteries and veins) which have a smooth muscle layer and thus show true endomysial positivity (Fig.1, row Strong positives, images No.3,4,5).

Accordingly here are suggestions to the title
- Recognizing IgA-Class Coeliac Disease-specific Autoantibody Binding Patterns on Monkey Liver Substrate Through EfficientNet Architectures and Deep Learning
- Recognizing IgA-Class Endomysial Antibody Equivalent Binding Patterns on Monkey Liver Substrate Through EfficientNet Architectures and Deep Learning
- Recognizing IgA-Class Endomysial Binding Patterns on Monkey Liver Portal Vessels Through EfficientNet Architectures and Deep Learning…

If the authors choose the first way, I suggest the following changes in the introduction and methods, where the additional abbreviation can be introduced and used thereafter in a clear way.

Line 88: „Positive results typically appear as granular or reticulin-like patterns depending on the substrate”. Indirect IF is never granular in celiac disease, what it appears granular in the liver, highly probably represents scattered cross-sections of the reticulin network. The positivity on the muscularis mucosae is following the endomysium, not called reticulin. Please rephrase.

Line 117: Please do not use the term „EMA” here. I suggest to write „ by focusing
on monkey liver–based celiac autoantibody binding images, which can be regarded as diagnostically equivalent to EMA detected on classic muscle substrates, like monkey esophagus or human umbilical cord”.

Line 195 „classification of EMA images derived from monkey liver tissue” -please change to „classification of EMA equivalent images derived from monkey liver tissue”

Line 197 „Caetano dos Santos et al. (2019) explored machine learning techniques
for diagnosing celiac disease. They developed an automated method for classifying IgA-class EMA test images using Support Vector Machines (SVMs) „ - please change to „Caetano dos Santos et al. (2019) explored machine learning techniques for diagnosing celiac disease. They developed an automated method for classifying IgA-class EMA test images from human umbilical cord using Support Vector Machines (SVMs)”

Line 243 „This pattern is considered analogous to the endomysial staining seen in striated muscle substrates and supports the use of monkey liver as a valid alternative for EMA detection.”
Please change to „This pattern, called originally as reticulin antibody binding (Seah et al, 1971) is considered analogous to the endomysial staining seen in striated/smooth muscle substrates and supports the use of monkey liver as a valid alternative for celiac autoantibody detection, and thus for the classic EMA testing. Given the different image appearences, we shall call in the followings the positive celiac pattern on liver as EMA equivalent (EMAeq and IgA-EMAeq) as a clear distinction from the positivity on muscle endomysium.” The liver pattern abbreviations also can be chosen differently by the authors, but it should be clearly different from the conventional EMA abbreviation and used consistently in the results and discussion sections.

[Seah PP, Fry L, Rossiter MA, Hoffbrand AV, Holborow EJ. Anti-reticulin antibodies in childhood coeliac disease. Lancet. 1971 Sep 25;2(7726):681-2. doi: 10.1016/s0140-6736(71)92248-3.ű]

Line 249 „All EMA slides” please change to „All stained liver slides…”
Line 284 Table 1: please omit EMA from the first column, or replace with EMA-eq, just telling 2-Class, 3-Class, 4-Class evaluation are also sufficient

2) Figures 3,4,5. Some improvements were made, but still the white numbers on the yellow and light green backgrounds are not legible. I suggest to use black fonts for these fields.

Experimental design

Please provide the number of images obtained per specimen or per section, and the total number of original (not transformed) images used for the training.
If only one image was taken, please mention whether it was a simple part of the specimen or a composite (panoramic) one.

How many images from the same specimen were used for the evaluation of individual serum samples ?

Validity of the findings

Consider to omit to show the comparative results on the liver sections performed by the Caetano Dos Santos methods.
They used totally different substrate (human umbilical cord) and images, no conclusion can be made without seeing the pictures.
It is sufficient to mention this simulation in the discussion, but with pointing out that they used classic EMA substrate and you used EMA equivalent substrate. This important information is still missing from the current manuscript.

Reviewer 3 ·

Basic reporting

no comment

Experimental design

no comment

Validity of the findings

no comment

Additional comments

I am happy with the changes made to the manuscript and can recommend acceptance. All comments were addressed: most were implemented and for the remaining comments the authors gave a good justification for why they weren’t.

The results in Table 7 are a good addition to the manuscript and the introduction in particular has been improved significantly.

One final point: I understand why the authors only picked cases where both evaluators agreed (and it’s good that they highlight this in the revised version), however, it would have been interesting to also focus on the contentious cases as well in a separate analysis.

---

## Round 0.3 · accepted · Accept

Congratulations on your contribution to our journal!

Reviewer 2 ·

Basic reporting

The manuscript has substantially been improved.

Experimental design

ok

Validity of the findings

ok

Additional comments

I only have two minor comments

1. Please spell out EMA and EMA-eq abbreviation at the first time they appear in the introduction. Although these are there in the abstract, it would be more understandable.

2. Please check Figure 1. The 5th image in the second (negative) row looks having clear sinusoidal positivity. Was that regarded as negatve ?